# Inattentional blindness in anesthesiology: A gorilla is worth one thousand words

**Alessandro De Cassai**[1]*, **Sebastiano Negro**[2], **Federico Geraldini**[1], **Annalisa Boscolo**[1], **Nicolò Sella**[2], **Marina Munari**[1], **Paolo Navalesi**[1,2]

1 Anesthesia and Intensive Care Unit, University Hospital of Padua, Padua, Italy, 2 Department of Medicine, University of Padua, Padua, Italy

* alessandro.decassai@aopd.veneto.it

**Data Availability Statement:** All relevant data are within the paper and its Supporting information files.

## Abstract

### Introduction

People are not able to anticipate unexpected events. Inattentional blindness is demonstrated to happen not only in naïve observers engaged in an unfamiliar task but also in field experts with years of training. Anaesthesia is the perfect example of a discipline which requires a high level of attention and our aim was to evaluate if inattentional blindness can affect anesthesiologists during their daily activities.

### Materials and methods

An online survey was distributed on Facebook between May 1, 2021 and May 31, 2021.

The survey consisted of five simulated cases with questions investigating the anesthetic management of day-case surgeries. Each case had an introduction, a chest radiography, an electrocardiogram, preoperative blood testing and the last case had a gorilla embedded in the chest radiography.

### Results

In total 699 respondents from 17 different countries were finally included in the analysis.

The main outcome was to assess the incidence of inattentional blindness.

Only 34 (4.9%) respondents were able to spot the gorilla. No differences were found between anesthesiologists or residents, private or public hospitals, or between medical doctors with different experience.

### Discussion

Our findings assess that inattentional blindness is common in anesthesia, and ever-growing attention is deemed necessary to improve patient safety; to achieve this objective several strategies should be adopted such as an increased use of standardized protocols, promoting automation based strategies to reduce human error when performing repetitive tasks and discouraging evaluation of multiple consecutive patients in the same work shifts independently of the associated complexity.

**Funding:** The authors received no specific funding for this work.

**Competing interests:** The authors have declared that no competing interests exist.

## Introduction

People are not able to anticipate unexpected events as, by definition, they are unexpected. Despite it may appear tautological and rather ironic, this phenomenon, known as inattentional blindness, has been extensively studied in the past decades since it has several tangible consequences in daily-life, such as policemen not noticing a gun during a vehicle stop [1] or an audience not noticing a gorilla on the stage [2]. In the latter example, half of the viewers asked to count ball-passes in a short video were unable to detect a man with a gorilla suit entering the stage, turning to face the camera, thumping its chest, and exiting after a total of 9 seconds on the screen [3]. Inattentional blindness has been demonstrated to happen not only in naïve observers engaged in an unfamiliar task, but also in field experts with years of training [4].

Anaesthesia is a complex discipline which requires a high level of attention and is characterized by periods of intense activity while managing multiple tasks at the same time and periods with a lower workload demand. The lower workload periods may occupy substantial parts of the routine cases (up to 40%) being associated with lower cognitive and physical demand [5]. However, during the intense activity phase, the anesthesiologist has to pay different types of attention to ensure patient safety. The required attention types are focused (e.g.vital signs alarms), divided (e.g. evaluating at the same time patient's vital parameters, surgical field and the equipment such as ventilator at the same time), selective (e.g. selecting and choosing the important input from the multiple parameter monitor suppressing irrelevant or distracting information) and sustained (e.g. patient undergoing emergent major surgery for several hours and requiring hemodynamic and respiratory management throughout the duration of the surgery). Necessarily this field has seen a continuous technological evolution and a progressive introduction of practice standards over the last decades with a significant improvement in patient safety [6]. However, complications still happen, and while many of them are probably unavoidable, an analysis of anesthesia claims highlighted that about half of the anesthesia adverse events were preventable with additional monitoring and attentive vigilance [7].

The aim of our study was to evaluate if inattentional blindness can affect anesthesiologists during their daily activities, in particular whether anesthesiologists are able to identify an unexpected stimulus during routine preoperative examination.

## Methods

### Survey design

We conducted an online survey using the Google Form web-instrument (https://www.google.it/intl/it/forms/about/, Google, Mountain View, California, United States). The survey protocol was examined by the Institutional Review Board of Padua (Chairman: Dott. Sergi, reference 32148/2021), and a waiver of formal approval was granted considering the nature of the study.

Clear guidelines to design a survey do not exist, however, we followed general recommendations to design and conduct high-quality surveys [8, 9].

A member of the research team performed a bibliographic search in order to identify validated surveys on inattentional blindness in anesthesiology and the terms "inattentional blindness", "survey", "anesthesiology", "anesthesiologist" were used in different combinations on the following literature database: Scopus, Pubmed, CENTRAL.

However, no pre-existing surveys or survey questions were found.

Therefore, we designed the survey items following Peterson's acronym BRUSO: brief, relevant, unambiguous, specific, and objective [10].

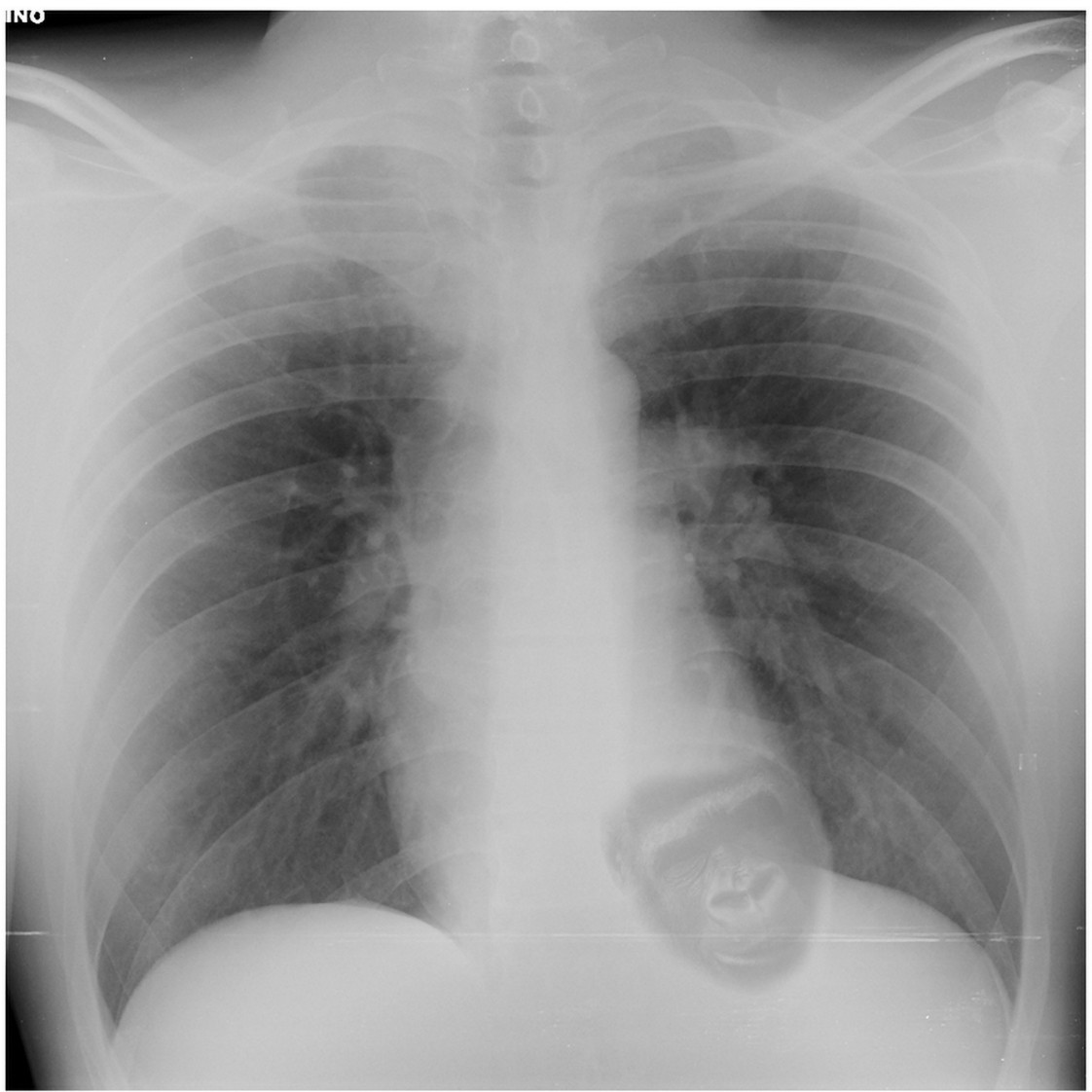

**Fig 1. Chest radiography used in the survey with the embedded gorilla in the heart.** The gorilla embedded in the heart is reprinted from https://commons.wikimedia.org/wiki/File:Female_Western_Lowland_Gorilla.jpg] and is of public domain. The figure is similar but not identical to the original image and is therefore for illustrative purposes only.

In order to achieve this objective we built our questionnaire keeping both the questionnaire and each question as short as possible in order to reduce the rate of partially completed questionnaires, we tried to avoid misleading or ambiguous terms and we asked only relevant questions.

The survey consisted of 5 simulated cases with single-choice, multiple-choice, and open-ended questions investigating the anesthetic management of day-case surgeries. Each case had an introduction (medical history, medications, proposed surgery), a chest radiography, an electrocardiogram and preoperative blood testing. Laboratory tests, chest radiography and electrocardiogram were the same for each case, without pathological findings. However, in the last case, a gorilla face was embedded inside the cardiac shadow (Fig 1).

Each case proposed a healthy patient or one with minor health problems undergoing non-major general surgery which did not present any particular concern or challenge related to the anesthetic management. Given the above, we would like to highlight that there cannot be a universally accepted correct answer by the respondents (e.g. spinal anesthesia compared to general anesthesia).

We decided to focus our survey on these simple cases in order to both increase respondent confidence with patient management irrespective of the personal experience gained as anesthesiologists and to investigate inattentional blindness during routine work.

For each case, participants were asked to choose the best anesthesia technique and to highlight remarkable findings in the case. For the main objective of this study, the question "Any comment on Patient 5?" was used to calculate the gorilla detection rate. However, we decided to consider the identification successful also if the physician incorrectly named the animal or labelled the cardiac shadow in the chest radiograph as "not normal".

The full list of questions is available for consultation (S1 File).

### Pretesting

Survey underwent four stages of pretesting. First, the survey was reviewed by the research team to evaluate the developed survey with particular attention to the questions to avoid "skip" or "branch" logic. Second, the survey was forwarded to thirty anesthesiology residents at the Padua University Hospital asking to evaluate text fluency and to report any typos and time required to complete the survey. In this phase, the time required to complete the survey was estimated to be lower than five minutes for all the participants. Third, in order to investigate the visibility of the embedded gorilla in the chest radiography we showed the embedded image to an audience of twenty residents. The image had an initial transparency of 100% (gorilla not visible at all), then the transparency was progressively lowered by 1%. At 15% all participants reported the gorilla. Fourth, hypothesizing that responders could use the mobile phone to respond to the survey, we assessed its visibility on different mobile phone screens. Twenty residents, uninvolved in the previous pre-testing phase, were able to clearly see the gorilla on phones with screens between 4.7 to 6.5 inches. All residents were able to identify the gorilla both on 4.7 and 6.5 inches screens.

### Population of interest and survey distribution plan

Population of interest are worldwide anesthesiologists.

We spread the survey on different groups for anesthesiologists on Facebook (Facebook, Inc., Menlo Park, California, United States) retrieved by searching groups using the following keywords: "anesthesia", "anaesthesia", "anesthesiology", "anaesthesiology", "critical care", "ICU".

We spread an invite to the survey on 1st May 2021 and we concluded collecting responses on the 31 May 2021.

A reminder was sent a week after the initial invitation.

Participants were invited to forward the survey to other anesthesiology groups and even to colleagues outside the Facebook platform.

### Statistical analysis

Variables are expressed as percentages and compared between groups using the chi-square test or Fisher's exact test when appropriate.

All statistical analyses were conducted using R version 3.4.0 (2017-04-21), statistical significance was set at $p$-values $<0.05$.

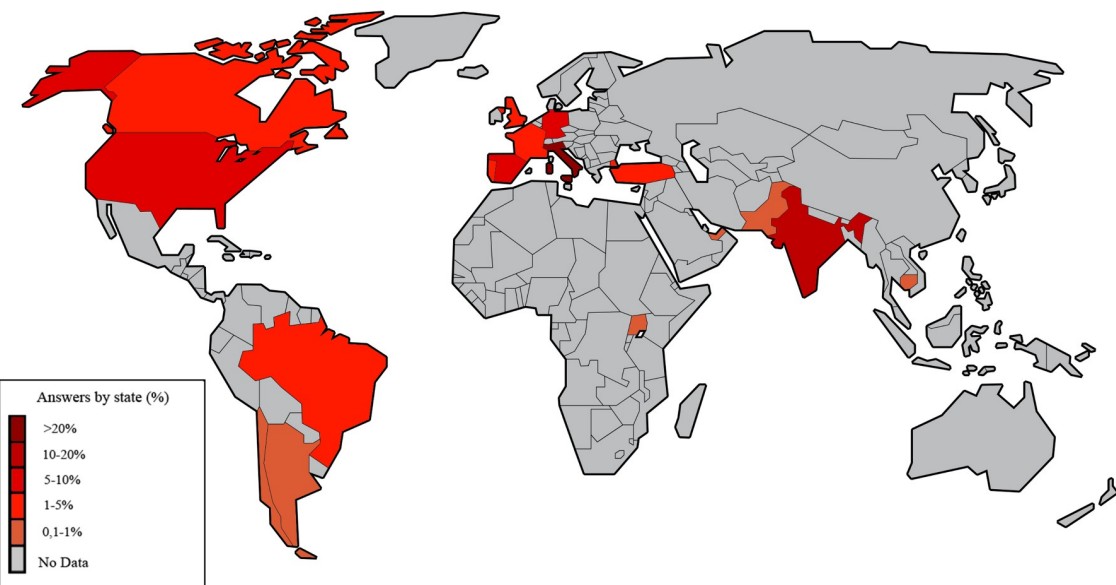

**Fig 2. Distribution of respondents.** World map is reprinted from https://commons.wikimedia.org/wiki/File:Simple_world_map_edit.svg under a CC BY 4.0 license, with permission from "Wikimedia Commons, the free media repository", original copyright 2019.

## Results

In total, 734 respondents completed the form. Of them, 35 were neither anesthesiologists or resident in anesthesia and therefore were excluded from the analysis leaving 699 respondents in the final analysis. We received completed forms from 17 countries and the geographical distribution of the centers is depicted in figure (Fig 2).

Characteristics of respondents are shown in Table 1, while raw data are available as S2 File. In our survey, only 34 (4.9%) respondents were able to spot the gorilla.

**Table 1. Respondents characteristics.**

|  | Total | Spotted | Not Spotted | p-value |
|---|---|---|---|---|
|  | (n = 699) | (n = 34) | (n = 665) |  |
| Anesthesiologist | 547 | 23 (4.2%) | 524 (95.8%) | 0.124 |
| Resident | 152 | 11 (7.2%) | 141 (92.8%) |  |
| Private Hospital | 116 | 2 (1.7%) | 114 (98.3%) | 0.107 |
| Public non-teaching Hospital | 258 | 11 (4.3%) | 247 (95.7%) |  |
| Public teaching Hospital | 325 | 21 (6.5%) | 304 (93.5%) |  |
| Exp:Resident | 152 | 11 (7.2%) | 141 (92.8%) | 0.396 |
| Exp: 0 to 5 years | 168 | 9 (5.3%) | 159 (94.7%) |  |
| Exp: 6 to 10 years | 111 | 6 (5.4%) | 105 (94.6%) |  |
| Exp: 11 to 15 years | 194 | 6 (3.1%) | 188 (96.9%) |  |
| Exp: 16 or more years | 74 | 2 (2.7%) | 72 (97.3%) |  |

Exp: Experience as anesthesiologists in years

## Discussion

Our work demonstrates that inattentional blindness exists in the anesthesia field and may be far more common than generally thought.

In our study anesthesiologists were asked to plan anesthesia management for five simulated cases. The proposed cases did not pose any particular challenge to the respondents because patients were healthy or with only minor health issues. Most patients undergoing anesthesia for minor surgery do not pose particular concerns to anesthesiologists, however the repetitive task of evaluating these patients could lead to lower attention threshold if the anesthesiologist is required to evaluate several of these patients consecutively in a brief time.

Contrary to common belief, perception is not a simple task where all the elements available to our senses are easily recognized and correctly interpreted. Indeed, for information to be correctly elaborated, it has to be perceived, understood, and finally analyzed for its possible future implications. This concept represents the three levels of situational awareness, defined as "the *perception* of elements of the environment within a volume of time and space, the *comprehension* of their meaning and the *projection* of their status in the near future" [11]. However, in certain situations, even being able to correctly gather and elaborate important information can potentially not be enough. For example, while working in a team, the shared situational awareness requires a common understanding of the situation as well as organized and timely communication. Ideally information is gathered, elaborated and shared within the team without delay, but actually this process is subject to potential failure on any level of the situational awareness chain [11]. Incorrectly gathered information could result from both a lack of detectability (i.e., physical barrier, equipment failure) or the inability to process the available information for various reasons.

Expanding the concept of inattentional blindness, the broader term of "inattentional insensitivity", defined as *"the inability to detect the sensation of a salient stimulus while performing a task within a congruent sensory modality"* [12], warrants consideration and can be applied to both medical and non medical situations. Undetected salient stimuli can be visual, as shown in this article, but the same phenomenon has been recognized for auditory [13] and tactile [12] stimuli. Inattentional blindness is a reality and it is not something to be ashamed of, because it is part of human being and it is not exclusive to a particular professional environment. Several examples of inattentional blindness are available in both medical (gynecology [14], general surgery [15] and cardiac surgery [16]) and non medical literature (driving vehicles [17] and aviation [18]).

This is not the first study evaluating inattentional blindness in anesthesiologists, with Ho et al [19] demonstrating that anesthesiologists were able to detect hypotension (90%) but not central venous line disconnection (23%) or head movement (42%) while watching a video of a major abdominal surgery. In our survey, the percentage of responders able to identify the unexpected event was even lower (4.9%) than in Ho et al. However, while in Ho's paper [19] participants were directly asked to search for abnormalities and these abnormalities are realistic and plausible in clinical practice (head movement and central venous catheter disconnection), we focused participants' attention on another task (preoperative evaluation) because it is a sine qua non for inattentional blindness. Moreover our stimulus was not predictable.

In our study anesthesiologists with more years of experience were equally likely to spot the gorilla compared to younger colleagues, apparently in contrast with Graham and Burke's study, which found that blindness for an unexpected event is greater for older adults than younger ones [20]. However, we did not directly investigate the age of the participants, but only their years of experience as anesthesiologists. While a correlation between age and years of experience surely exists, it represents only a surrogate of age, leaving the possibility of young

anesthesiologists with more years of experience or older anesthesiologists with just few years of experience. Moreover, other potential confounders exist, such as years of residency, and different digital support used to respond to the survey (mobile phone vs pc).

In anesthesia practice inattentional blindness deserves particular attention because it could have important safety implications, as an example, it has been reported that during anesthesia administration drug recording error is as high as one every eight administration [21], with error frequency increasing with a higher cognitive load [22]. Therefore, awareness of this phenomenon is of paramount importance to limit its dangerous consequences and guarantee patient safety. Anesthesia practice has made great strides in this direction, such as double checks before drugs administration is a common good clinical practices to avoid similarly looking vials or ampoule, implementation of checklists prior to any medical or surgical procedure reducing the human component, gas cylinder having a particular pin configuration for each medical gas on the yoke of the anaesthesia machine in order to avoid gas cylinder exchange [22] and so on.

Our study has some limitations that we would like to discuss. First, we do not know the total number of anesthesiologists that received the invite to complete the survey and this prevents us from calculating non respondent rate and to model their characteristics using a wave analysis. For this reason, while the aim of our paper to investigate the presence of inattentional blindness in anesthesiology practice was accomplished, we recognize that the true incidence could have been over- or under-estimated.

Second, the time in which the anesthesiologist looked at the image was not standardized. It could be argued that the gorilla would eventually be noticed with more visualization time. However, it has been proved that even though subjects have more opportunity to detect the unexpected object the longer it remains on-screen, the vast majority of noticing events occur in the first 1.5 seconds or not at all [23].

Third, we recognize that given the nature of the study (a survey distributed via social network) it is not possible to determine how seriously the respondents assessed each case.

## Conclusions

Inattentional blindness is a common and unavoidable phenomenon in anesthesia. An ever-growing attention is deemed necessary to improve patient safety; to achieve this objective several strategies should be adopted such as an increased use of standardized protocols, promoting automation based strategies to reduce human error when performing repetitive tasks and discouraging evaluation of multiple consecutive patients in the same work shifts independently of the associated complexity.

## Supporting information

**S1 File. Survey questionnaire.**
(PDF)

**S2 File. Dataset.**
(XLSX)

## Author Contributions

**Conceptualization:** Alessandro De Cassai, Sebastiano Negro, Federico Geraldini.

**Data curation:** Alessandro De Cassai.

**Formal analysis:** Alessandro De Cassai.

**Investigation:** Alessandro De Cassai.

**Methodology:** Alessandro De Cassai.

**Supervision:** Alessandro De Cassai, Paolo Navalesi.

**Writing – original draft:** Alessandro De Cassai, Sebastiano Negro, Federico Geraldini, Annalisa Boscolo, Nicolò Sella, Marina Munari, Paolo Navalesi.

**Writing – review & editing:** Alessandro De Cassai, Sebastiano Negro, Federico Geraldini, Annalisa Boscolo, Nicolò Sella, Marina Munari, Paolo Navalesi.

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
