## [Decision Letter · Decision Letter 0]

13 Jul 2021

PONE-D-21-20456

Inattentional blindness in Anesthesiology: a gorilla is worth one thousand words.

PLOS ONE

Dear Dr. De Cassai,

Thank you for submitting your manuscript to PLOS ONE. After careful consideration, we feel that it has merit but does not fully meet PLOS ONE’s publication criteria as it currently stands. Therefore, we invite you to submit a revised version of the manuscript that addresses the points raised during the review process.

We look forward to receiving your revised manuscript.

Kind regards,

Dylan A Mordaunt

Academic Editor

PLOS ONE

Journal Requirements:

2. PLOS ONE does not copy edit accepted manuscripts (https://journals.plos.org/plosone/s/criteria-for-publication#loc-5). To that effect, please ensure that your submission is free of typos and grammatical errors. 

3. We note that Figure 1 in your submission contain copyrighted images. All PLOS content is published under the Creative Commons Attribution License (CC BY 4.0), which means that the manuscript, images, and Supporting Information files will be freely available online, and any third party is permitted to access, download, copy, distribute, and use these materials in any way, even commercially, with proper attribution. For more information, see our copyright guidelines: http://journals.plos.org/plosone/s/licenses-and-copyright.

4. We note that Figure 2 in your submission contain [map/satellite] images which may be copyrighted. All PLOS content is published under the Creative Commons Attribution License (CC BY 4.0), which means that the manuscript, images, and Supporting Information files will be freely available online, and any third party is permitted to access, download, copy, distribute, and use these materials in any way, even commercially, with proper attribution. For these reasons, we cannot publish previously copyrighted maps or satellite images created using proprietary data, such as Google software (Google Maps, Street View, and Earth). For more information, see our copyright guidelines: http://journals.plos.org/plosone/s/licenses-and-copyright.

Additional Editor Comments (if provided):

I agree with the reviewer, this study injects some humour into a serious issue and adds novelty in a similar way to Christmas BMJ. I intend to accept this submission but I would like the reviewers to go away and expand the discussion to include semantically similar concepts to inattentional blindness in their discussion. They have cited the one previous simulation study but for the authors' own benefit and for the benefit of increased future impact, I think they could expand to discuss similar/relevant concepts such as situational awareness, as they have been studies in anaesthetics, peri-operative or critical care circumstances. Have some examples as follows- https://blog.oup.com/2015/01/perceptual-errors-inattenional-blindness/ and https://pubs.asahq.org/anesthesiology/article/118/3/729/13561/Situation-Awareness-in-AnesthesiaConcept-and (NB: these are not self-references). Situational blindness is a critical concept in human factors studies and I also think it's something you'll find has highly relevant military and aviation links. I look forward to reading the response, the intention is not to get you to completely rework the paper, just to consider adding some generalisability and additional value. Kind regards,

Reviewers' comments:

Reviewer's Responses to Questions

**Comments to the Author**

1. Is the manuscript technically sound, and do the data support the conclusions?

Reviewer #1: Yes

2. Has the statistical analysis been performed appropriately and rigorously? 

Reviewer #1: Yes

3. Have the authors made all data underlying the findings in their manuscript fully available?

Reviewer #1: Yes

4. Is the manuscript presented in an intelligible fashion and written in standard English?

Reviewer #1: Yes

5. Review Comments to the Author

Reviewer #1: The authors present a wonderful piece of research: they raise attention for a very serious topic with a highly entertaining method, and thus succeed in making the reader both think and smile.

Inattentional blindness is, as pointed out in their discussion, a major topic for health care providers, especially in but definitely not limited to perioperative care. The authors confronted anaesthesiologists with medical cases, in which a gorilla face was hidden in a chest-x-ray. Only one third spotted the gorilla.

Of course, the study can be criticized, as biases in the selection of the sample and the response rates are not completely transparent, (as the authors mention in their discussion) but this does not weaken the fundamental statement that a large proportion of experienced clinicians were not aware of the gross conspicuity.

I congratulate the authors for their work! Thank you!

6. PLOS authors have the option to publish the peer review history of their article (what does this mean?). If published, this will include your full peer review and any attached files.

Reviewer #1: No

---

## [Author Response · Author response to Decision Letter 0]

18 Jul 2021

Padova 18/07/2021

Dear Editor, 

Dear Reviewer,

We are submitting a revised version of our manuscript.

We hope that our manuscript meets the high standards of PLOS ONE and it is evaluable for publication!

Thank you for your consideration!

Alessandro De Cassai on behalf of all coauthors

Editor

We modified manuscript accordingly

2. PLOS ONE does not copy edit accepted manuscripts (https://journals.plos.org/plosone/s/criteria-for-publication#loc-5). To that effect, please ensure that your submission is free of typos and grammatical errors. 

We read the manuscript a further time removing some typos. Thank you for pointing it out.

3. We note that Figure 1 in your submission contain copyrighted images. All PLOS content is published under the Creative Commons Attribution License (CC BY 4.0), which means that the manuscript, images, and Supporting Information files will be freely available online, and any third party is permitted to access, download, copy, distribute, and use these materials in any way, even commercially, with proper attribution. For more information, see our copyright guidelines: http://journals.plos.org/plosone/s/licenses-and-copyright.

We were unable to obtain permission from the original copyright holder to publish under CC BY 4.0. For these reason we supply a replacement adding in the figure legend that the figure is similar but not identical to the original image and is therefore for illustrative purposes only.

4. We note that Figure 2 in your submission contain [map/satellite] images which may be copyrighted. All PLOS content is published under the Creative Commons Attribution License (CC BY 4.0), which means that the manuscript, images, and Supporting Information files will be freely available online, and any third party is permitted to access, download, copy, distribute, and use these materials in any way, even commercially, with proper attribution. For these reasons, we cannot publish previously copyrighted maps or satellite images created using proprietary data, such as Google software (Google Maps, Street View, and Earth). For more information, see our copyright guidelines: http://journals.plos.org/plosone/s/licenses-and-copyright.

We are not sure who is the original copyright holder of the map we used. For this reason we recreate the map starting from an image with CC BY 4-0 license

Done

We reviewed the references, however, we were not able to see any retracted article among the articles we cited. However, if you believe some of papers we have cited has been retracted please let us know so we could remove it from reference list.

7. I agree with the reviewer, this study injects some humour into a serious issue and adds novelty in a similar way to Christmas BMJ. I intend to accept this submission but I would like the reviewers to go away and expand the discussion to include semantically similar concepts to inattentional blindness in their discussion. They have cited the one previous simulation study but for the authors' own benefit and for the benefit of increased future impact, I think they could expand to discuss similar/relevant concepts such as situational awareness, as they have been studies in anaesthetics, peri-operative or critical care circumstances. Have some examples as follows- https://blog.oup.com/2015/01/perceptual-errors-inattenional-blindness/ and https://pubs.asahq.org/anesthesiology/article/118/3/729/13561/Situation-Awareness-in-AnesthesiaConcept-and (NB: these are not self-references). Situational blindness is a critical concept in human factors studies and I also think it's something you'll find has highly relevant military and aviation links. I look forward to reading the response, the intention is not to get you to completely rework the paper, just to consider adding some generalisability and additional value. Kind regards,

We tried to expand the discussion adding value to the overall manuscript. Thank you for your comment.

Reviewers' comments:

Comments to the Author

Reviewer #1: The authors present a wonderful piece of research: they raise attention for a very serious topic with a highly entertaining method, and thus succeed in making the reader both think and smile.

Inattentional blindness is, as pointed out in their discussion, a major topic for health care providers, especially in but definitely not limited to perioperative care. The authors confronted anaesthesiologists with medical cases, in which a gorilla face was hidden in a chest-x-ray. Only one third spotted the gorilla.

Of course, the study can be criticized, as biases in the selection of the sample and the response rates are not completely transparent, (as the authors mention in their discussion) but this does not weaken the fundamental statement that a large proportion of experienced clinicians were not aware of the gross conspicuity. I congratulate the authors for their work! Thank you!

We would like to thank the reviewer for his kind words

---

## [Decision Letter · Decision Letter 1]

31 Aug 2021

PONE-D-21-20456R1

Inattentional blindness in anesthesiology: a gorilla is worth one thousand words.

PLOS ONE

Dear Dr. De Cassai,

Thank you for submitting your manuscript to PLOS ONE. After careful consideration, we feel that it has merit but does not fully meet PLOS ONE’s publication criteria as it currently stands. Therefore, we invite you to submit a revised version of the manuscript that addresses the points raised during the review process.

We look forward to receiving your revised manuscript.

Kind regards,

Dylan A Mordaunt

Academic Editor

PLOS ONE

Journal Requirements:

Additional Editor Comments (if provided):

Although human factors psychology is an important part of healthcare safety sciences, and in particular areas like anaesthesia, there were relatively few individuals available to review the manuscrupt. With this in mind, I sought input from an academic specialist in human factors psychology. This expert's recommendation was to reject the article on the basis of what you can see in the reviewer's responses. The first issue was a perceived lack of new findings- whilst the suggestion is that this is similar to previous work, I think the difference is that it's applied to this specific scenario (i.e. anaesthetic safety) and whilst the approach may be similar, it doesn't appear to be either direct duplication nor replication of previous work- so though perhaps iterative or redundant from a human factors psychology perspective, I think it offers value to the health safety sciences community. With that in mind, both reviewer 2 and 3's comments should be considered as I think they have the potential to add value to the manuscript. I will leave it to the authors to decide which they agree with and to respond to those which they don't, but they do increase the bar for re-review to major. It may be worth considering whether the authors make greater reference to the phenomena of inattentional blindness in other fields, and the wider implications for other areas of healthcare.

Reviewers' comments:

Reviewer's Responses to Questions

**Comments to the Author**

1. If the authors have adequately addressed your comments raised in a previous round of review and you feel that this manuscript is now acceptable for publication, you may indicate that here to bypass the “Comments to the Author” section, enter your conflict of interest statement in the “Confidential to Editor” section, and submit your "Accept" recommendation.

Reviewer #1: All comments have been addressed

Reviewer #2: (No Response)

Reviewer #3: (No Response)

2. Is the manuscript technically sound, and do the data support the conclusions?

Reviewer #1: Yes

Reviewer #2: Partly

Reviewer #3: Yes

3. Has the statistical analysis been performed appropriately and rigorously? 

Reviewer #1: Yes

Reviewer #2: No

Reviewer #3: Yes

4. Have the authors made all data underlying the findings in their manuscript fully available?

Reviewer #1: Yes

Reviewer #2: Yes

Reviewer #3: Yes

5. Is the manuscript presented in an intelligible fashion and written in standard English?

Reviewer #1: Yes

Reviewer #2: No

Reviewer #3: Yes

6. Review Comments to the Author

Reviewer #1: The authors present a revised version of their manuscript. They included satisfying responses all issues raised.

Reviewer #2: The manuscript is incomplete and not ready for the review.

1. I don't see any new findings in this study.

2. Instead of a general remark (ever-growing attention is deemed necessary to improve patient safety.), more specific suggestions should be provided.

3. This study analyzed only incidence rates. How about diagnostic accuracy for each case?

4. Each section should consist of paragraphs, but some sections do not have paragraphs. In addition, some sentences have grammatical errors.

5. The conclusion consists of only one sentence.

When simulated tasks are used, it is essential to ensure their validity. Discuss their validity.

1. Are simulated tasks valid in terms of image size, resolution, visual angle, task time, etc?

2. How seriously would the respondent consider each case?

3. How confidently would the respondent answer each question?

4. How similar were the five cases to the actual cases in terms of workload?

For each simulated case, explain in detail where to look at for a typical diagnosis.

Describe how the developed survey meets the BRUSO criteria.

If laboratory tests, chest radiography and electrocardiogram are the same for the first several cases, without pathological findings, it is very unlikely to pay attention to these data for the remaining cases, thus more likely to miss the gorilla.

How did you conclude whether the gorilla was (was not) spotted by respondents? I don't see a direct question on this.

The abstract should be rewritten.

1. The abstract should be one paragraph without any subsections.

2. An online survey was spread on Facebook (Facebook, Inc., Menlo Park, California, United States) between May 1, 2021 and May 31, 2021.

: Too much information is provided.

3. The survey consisted of simulated cases.

: More information is needed. (The survey consisted of five simulated cases.)

4. There are no meaningful findings/implications.

In the introduction, the nature of anaesthesia-related tasks should be described in detail for readers without medical information. For example, what types of attention (focused, selective, divided, and sustained) are relevant to anaesthesia-related tasks ? How many and what data channels are involved? and what is a typical procedure involved in diagnosis?

The statistical analysis section should be described more in detail.

Reviewer #3: Inattentional blindness is a serious concern that is somewhat lacking attention in the public and in serious workplaces. I agree with the previous reviewer and editor, the authors do a good job bringing light to this topic using humility that will attract a wide audience to a serious topic. I also agree that there are some pitfalls to this paper and study; however, I agree with the previous reviewer that they do not affect the overall statement of the paper.

Despite that, I do have a few minor comments regarding grammar:

1) Although not significant, the rate at which the gorilla was spotted did trend towards decreasing with experience. Please be sure to mention this in your discussion in the topic regarding age and inattentional blindness.

2) Abstract, Materials and Methods - please replace "spreaded" with spread or distributed.

3) Page 7 (Methods, Population of Interest and Survey Distribution Plan - please revise "we initiated to spread the invite to respond to the survey" to "We spread an invite to the survey" or "We began distribution of the survey invitation" or equivalent

4) Page 8 (Results, paragraph 1) - there are two references to figure 2. Please remove one.

5) Page 11 (Discussion, sentence 1) - remove that from "exists in the anesthesia field and that may be far more common than generally thought"

6) Page 13 (discussion) - remove the new line following "our study has some limitations that we would like to discuss"

7) Page 14 (discussion) - reword "using as example a wave analysis model"

7. PLOS authors have the option to publish the peer review history of their article (what does this mean?). If published, this will include your full peer review and any attached files.

Reviewer #1: No

Reviewer #2: No

Reviewer #3: **Yes: **Matthew Scott Sherwood

---

## [Author Response · Author response to Decision Letter 1]

1 Sep 2021

Padova 01/09/2021

Dear Editor,

Dear Reviewers,

Thank you for your time and effort in improving our manuscript. We deeply appreciate it.

We tried to respond to your concerns and to modify the manuscript accordingly

We hope our manuscript is suitable for publication in your prestigious journal!

Alessandro De Cassai on behalf of all coauthors

Reviewer #1: The authors present a revised version of their manuscript. They included satisfying responses all issues raised.

Reviewer #1 We would like to thanks Reviewer#1 for his kind words.

 Reviewer #2:

 Q)Instead of a general remark (ever-growing attention is deemed necessary to improve patient safety.), more specific suggestions should be provided. The conclusion consists of only one sentence

A)Thank you for your comment. We added the following to the conclusion section:

“Inattentional blindness is a common and unavoidable phenomenon in anesthesia. An ever-growing attention is deemed necessary to improve patient safety; to achieve this objective several strategies should be adopted such as an increased use of standardized protocols, promoting automation based strategies to reduce human error when performing repetitive tasks and discouraging evaluation of multiple consecutive patients in the same work shifts independently of the associated complexity.”

Q)This study analyzed only incidence rates. How about diagnostic accuracy for each case? For each simulated case, explain in detail where to look at for a typical diagnosis.

A)Dear Reviewer, there was not a “correct answer”. We tried to explain this adding the following paragraph to the method section:

“Each case proposed a healthy patient or one with minor health problems undergoing non-major general surgery which did not present any particular concern or challenge related to the anesthetic management. Given the above, we would like to highlight that there cannot be a universally accepted correct answer by the respondents (e.g. spinal anesthesia compared to general anesthesia). 

We decided to focus our survey on these simple cases in order to both increase respondent confidence with patient management irrespective of the personal experience gained as anesthesiologists and to investigate inattentional blindness during routine work.”

Q)Each section should consist of paragraphs, but some sections do not have paragraphs. In addition, some sentences have grammatical errors.

A)Thank you for noticing it, we corrected some typos in the text.

If the reviewer thinks more paragraph titles are appropriate we kindly ask to indicate to us which one we should add. 

Q)When simulated tasks are used, it is essential to ensure their validity. Discuss their validity.

1. Are simulated tasks valid in terms of image size, resolution, visual angle, task time, etc?. How seriously would the respondent consider each case?. How confidently would the respondent answer each question?. How similar were the five cases to the actual cases in terms of workload?

A)Validity is discussed in the pretesting section of the manuscript:

In order to respond to the second part of your question we added the following to the method section

“Each case proposed a healthy patient or one with minor health problems undergoing non-major general surgery which did not present any particular concern or challenge related to the anesthetic management. Given the above, we would like to highlight that there cannot be a universally accepted correct answer by the respondents (e.g. spinal anesthesia compared to general anesthesia). 

We decided to focus our survey on these simple cases in order to both increase respondent confidence with patient management irrespective of the personal experience gained as anesthesiologists and to investigate inattentional blindness during routine work.”

Moreover, we added the following in the limitation section

Third, we recognize that given the nature of the study (a survey distributed via social network) it is not possible to determine how seriously the respondents assessed each case.

Describe how the developed survey meets the BRUSO criteria.

We added the following in the method section

“In order to achieve this objective we built our questionnaire keeping both the questionnaire and each question as short as possible in order to reduce the rate of partially completed questionnaires, we tried to avoid misleading or ambiguous terms and we asked only relevant questions.”

If laboratory tests, chest radiography and electrocardiogram are the same for the first several cases, without pathological findings, it is very unlikely to pay attention to these data for the remaining cases, thus more likely to miss the gorilla.

Thank you for pointing out this point. We added the following in the discussion section:

In our study anesthesiologists were asked to plan anesthesia management for five simulated cases. The proposed cases did not pose any particular challenge to the respondents because patients were healthy or with only minor health issues. Most patients undergoing anesthesia for minor surgery do not pose particular concerns to anesthesiologists, however the repetitive task of evaluating these patients could lead to lower attention threshold if the anesthesiologist is required to evaluate several of these patients consecutively in a brief time.

How did you conclude whether the gorilla was (was not) spotted by respondents? I don't see a direct question on this.

Added the following in the method section

“ For the main objective of this study, the question “Any comment on Patient 5?” was used to

calculate the gorilla detection rate. However, we decided to consider the identification successful also if the physician incorrectly named the animal or labelled the cardiac shadow in the chest radiograph as “not normal”.” 

Q)The abstract should be rewritten.

1. The abstract should be one paragraph without any subsections.

Dear Reviewer, in many PLOSone article subsections are used. We would like to keep the subsections, however, if editor request it we will remove it.

2. An online survey was spread on Facebook (Facebook, Inc., Menlo Park, California, United States) between May 1, 2021 and May 31, 2021: Too much information is provided.

Taken

3. The survey consisted of simulated cases.: More information is needed. (The survey consisted of five simulated cases.)

Taken

4. There are no meaningful findings/implications.

Taken

Q)In the introduction, the nature of anaesthesia-related tasks should be described in detail for readers without medical information. For example, what types of attention (focused, selective, divided, and sustained) are relevant to anaesthesia-related tasks ? How many and what data channels are involved? and what is a typical procedure involved in diagnosis?

Added the following in the introduction:

Anaesthesia is a complex discipline which requires a high level of attention and is characterized by periods of intense activity while managing multiple tasks at the same time and periods with a lower workload demand. The lower workload periods may occupy substantial parts of the routine cases (up to 40%) being associated with lower cognitive and physical demand [slagle]. However, during the intense activity phase, the anesthesiologist has to pay different types of attention to ensure patient safety. The required attention types are focused (e.g.vital signs alarms), divided (e.g. evaluating at the same time patient’s vital parameters, surgical field and the equipment such as ventilator at the same time), selective (e.g. selecting and choosing the important input from the multiple parameter monitor suppressing irrelevant or distracting information) and sustained (e.g. patient undergoing emergent major surgery for several hours and requiring hemodynamic and respiratory management throughout the duration of the surgery).

Q)The statistical analysis section should be described more in detail.

We recognize that the statistical analysis section is quite small. However, it is the statistical analysis we actually used. We do not believe that increasing the paragraph would be helpful to the readers to replicate our study.

Reviewer #3: Inattentional blindness is a serious concern that is somewhat lacking attention in the public and in serious workplaces. I agree with the previous reviewer and editor, the authors do a good job bringing light to this topic using humility that will attract a wide audience to a serious topic. I also agree that there are some pitfalls to this paper and study; however, I agree with the previous reviewer that they do not affect the overall statement of the paper.

We would like to thank the reviewer for his comment.

Despite that, I do have a few minor comments regarding grammar:

1) Although not significant, the rate at which the gorilla was spotted did trend towards decreasing with experience. Please be sure to mention this in your discussion in the topic regarding age and inattentional blindness.

Dear reviewer, we would be happy to find a statistical significance (because this finding would agree with the previous literature). However, it was not. We believe that stating that we “found a trend towards decrease with experience” would be not totally valid. And if reviewer and editor agree we would prefer to leave the sentence in its original form.

2) Abstract, Materials and Methods - please replace "spreaded" with spread or distributed.

Taken

3) Page 7 (Methods, Population of Interest and Survey Distribution Plan - please revise "we initiated to spread the invite to respond to the survey" to "We spread an invite to the survey" or "We began distribution of the survey invitation" or equivalent

Taken

4) Page 8 (Results, paragraph 1) - there are two references to figure 2. Please remove one.

Taken

5) Page 11 (Discussion, sentence 1) - remove that from "exists in the anesthesia field and that may be far more common than generally thought"

Taken

6) Page 13 (discussion) - remove the new line following "our study has some limitations that we would like to discuss"

Taken

7) Page 14 (discussion) - reword "using as example a wave analysis model"

Taken

---

## [Editor Report · Decision Letter 2]

3 Sep 2021

Inattentional blindness in anesthesiology: a gorilla is worth one thousand words.

PONE-D-21-20456R2

Dear Dr. De Cassai,

We’re pleased to inform you that your manuscript has been judged scientifically suitable for publication and will be formally accepted for publication once it meets all outstanding technical requirements.

Kind regards,

Dylan A Mordaunt

Academic Editor

PLOS ONE
---

## [Editor Report · Acceptance letter]

15 Sep 2021

PONE-D-21-20456R2 

Inattentional blindness in anesthesiology: a gorilla is worth one thousand words. 

Dear Dr. De Cassai:

I'm pleased to inform you that your manuscript has been deemed suitable for publication in PLOS ONE. Congratulations! Your manuscript is now with our production department. 

Kind regards, 

on behalf of

Dr. Dylan A Mordaunt 

Academic Editor

PLOS ONE